# ADAPTIVE CALIBRATION FOR FAIRER FACIAL RECOGNITION

## ABSTRACT

We introduce a novel calibration strategy for facial recognition, Adaptive Calibration, which maps cosine similarity between normalized embeddings to well-calibrated probabilities. By incorporating local embedding context into the calibration process, Adaptive Calibration is able to correct for the fact identical distances correspond to different match probabilities in different embedding regions. This yields improved calibration that adapts to local embedding distributions without requiring demographic metadata. Experiments with standard benchmarks for face verification across a variety of pretrained models demonstrate that our approach consistently dominates existing methods both on accuracy (AUROC) and fairness metrics. Our method provides a practical solution for more equitable facial recognition systems, without requiring demographic group annotations, and while improving overall performance. Unlike existing approaches that often rely on discrete clustering with additional hyperparameters or cause abrupt calibration shifts at cluster boundaries, our method provides continuous, region-specific calibration that avoids both the algorithmic limitations and the issue of "leveling down" whereby fairness is achieved by degrading performance for already disadvantaged groups.

## 1 INTRODUCTION

Face verification has found widespread adoption in various domains such as border security, surveillance, and law enforcement (Sepas-Moghaddam et al., 2019; Grother et al., 2019; Hill, 2024; Kotwal & Marcel, 2025). Despite these advances, modern systems exhibit substantial demographic biases. Error rates for these systems can be an order of magnitude higher for individuals with dark skin tones compared to those with lighter skin (Grother et al., 2019; Buolamwini & Gebru, 2018). Because of this, when law enforcement relies on biased facial recognition systems, the resulting misidentifications can result in wrongful arrests, predominantly affecting marginalized communities (Buolamwini & Gebru, 2018; Cavazos et al., 2020).

Unlike classification, which tends to be trained in a probabilistic framework (Prince, 2023), modern approaches to facial recognition are typically trained to learn optimal face embeddings through distance-based approaches such as contrastive learning (S, 2023). These models transform face images into high-dimensional feature vectors (embeddings), where the distance (cosine similarity) between normalized pairs of embeddings indicates identity similarity, with smaller distances suggesting the faces belong to the same person. Although these techniques are effective and have led to substantial progress in the field, the outputs of facial recognition systems do not map cleanly onto probabilities. Consider two pairs of faces with similar distances between members of each pair. While the distances are similar, the probability of each pair being a correct match can shift depending on where they lie in the embedding space (Salvador et al., 2022).

The errors associated with this fall disproportionately on minority groups underrepresented in the dataset (Grother et al., 2019; Cherepanova et al., 2022). This is unsurprising, as most facial recognition training sets, especially large-scale datasets used to train state-of-the-art models, are highly imbalanced (Wang et al., 2019; Robinson et al., 2020; Buolamwini & Gebru, 2018).

This imbalance is caused by minority groups occupying low-density regions of the embedding space, where the behavior of an ML system is less well-posed (Cherepanova et al., 2022). In these sparse regions, distances between embeddings become less reliable indicators of identity matching,

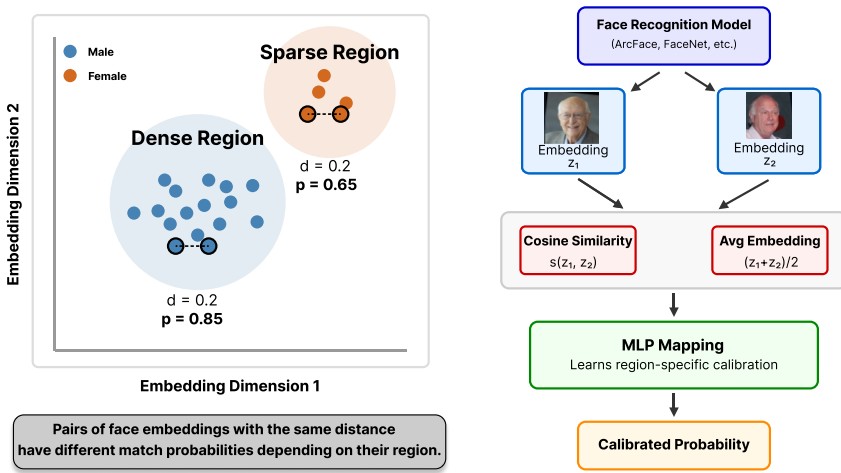

Figure 1: Adaptive Calibration Framework. Left: Identical distances in the embedding space yield different match probabilities in different regions of the space. Right: Our method uses both cosine similarity and average embedding location to learn region-specific calibrations through MLP/Linear mapping.

as the model has seen fewer examples to properly calibrate these relationships. Consequently, a global threshold that works well for majority groups often performs poorly for underrepresented demographics, leading to higher false rejection rates, if the threshold is too strict, or higher false acceptance rates, if the threshold is too lenient, for these groups (Huang et al., 2019).

Transforming these distances into calibrated probabilities (or equivalently adjusting distances in the original embedding space) is essential to ensure fair and accurate outputs across diverse groups by accounting for local embedding space variations (Salvador et al., 2022). How raw distances are converted significantly impacts system performance. Standard global approaches such as Platt scaling and beta calibration rely on a single monotonic mapping and, necessarily reproduce the same disparities seen when thresholding raw distances (Rahimi et al., 2020; Pleiss et al., 2017; Ding et al., 2021; Salvador et al., 2022).

To address this, researchers employ two primary strategies: model-level modifications and post-processing methods. Model-level approaches alter the training process or architecture to reduce bias in the embedding space, often at the expense of overall accuracy (Salvador et al., 2022; Conti et al., 2024; Kotwal & Marcel, 2025; Sohn et al., 2017; Kan et al., 2014; Serna et al., 2022; Wang et al., 2019).

Post-processing methods, applied after model training (post-hoc) (Salvador et al., 2022; Kotwal & Marcel, 2025; Dhar et al., 2021; Linghu et al., 2024; Dhar et al., 2020; Terhörst et al., 2020; Conti et al., 2024), adjust a trained model's outputs to reduce bias while seeking to maintain accuracy. These methods differ in approach: some directly generate calibrated probabilities, whereas others modify distance metrics, requiring an additional calibration step to produce probabilities. However, such adjustments often incur a cost, via "leveling down" (Mittelstadt et al., 2023), whereby fairness gains come at the cost of degrading performance for some groups.

To address these limitations, we introduce Adaptive Calibration, a novel approach that learns a continuous region-aware calibration function as seen in Figure 1. Our method leverages a multilayer perceptron (MLP) to integrate local embedding context with cosine similarity scores, producing well-calibrated match probabilities without explicit clustering. Operating as a post-hoc addition to any pretrained face recognition system, Adaptive Calibration can be integrated into existing pipelines. It achieves superior fairness–accuracy trade-offs, outperforming existing approaches as demonstrated

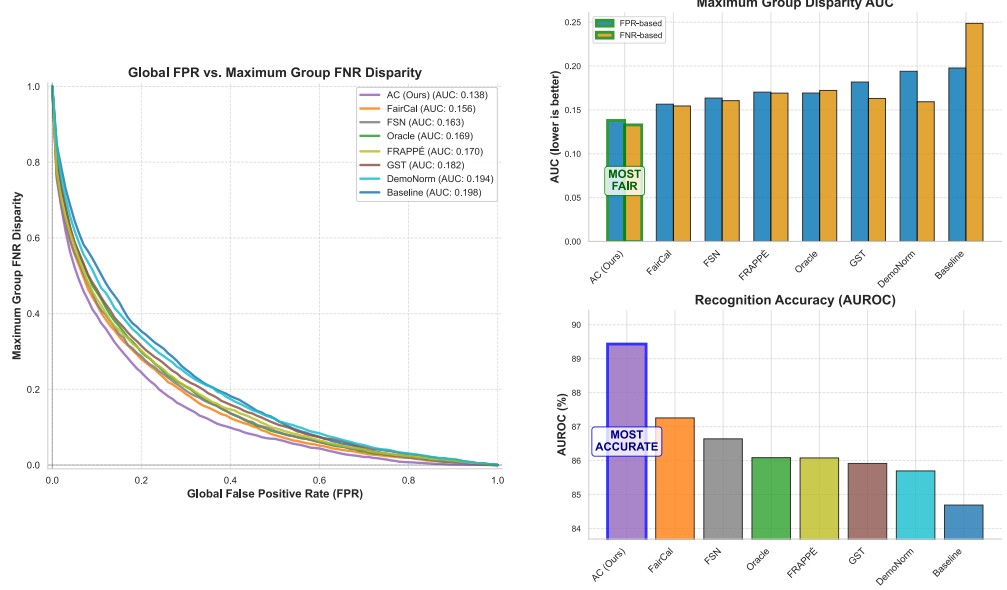

Figure 2: Fairness and Accuracy. Left: Global FPR vs. Max Group FNR curves. Here a lower AUC indicate better performance on the groups the algorithm works worst on. Top Right: we measure maximal discrepancy between per group AUC with respect to two curves. Global FPR vs. Group FNR; and global FNR vs. Group FPR - again lower is better. The bottom right panel shows recognition accuracy using AUROC (higher is better). Adaptive Calibration achieves the best fairness AUC and greatest accuracy. Results show the standard MLP Adaptive Calibration (AC) variant. The base model here is FaceNet trained on WebFace tested on RFW.

in Figure 2. All methods, evaluation criteria, and implementations will be open-source and can be seen in the supplementary materials. We hope this will help practitioners deploy more fair and accurate facial recognition system pipelines.

**Key Contributions:** We propose a novel method, adaptive calibration, that leverages local embedding context to dynamically map similarity scores into well-calibrated probabilities avoiding pitfalls that harm accuracy and fairness. We empirically demonstrate that our method dominates existing post-hoc calibration techniques, achieving superior fairness and accuracy. Additionally, we create a new benchmark for calibration testing by leveraging the DemogPairs (Hupont & Fernandez, 2019) dataset structure and will open-source the paired data for benchmark evaluations.

## 2 RELATED WORK

### 2.1 FACIAL RECOGNITION

Face recognition systems leverage deep neural networks to transform raw images into discriminative embeddings, enabling identity verification (Deng et al., 2022; S, 2023). As when these embeddings are L2-normalized, the cosine similarity serves as an effective metric for matching face pairs (Deng et al., 2022). This similarity score is typically compared against a threshold to determine whether two face images belong to the same identity. In practice, this threshold is calibrated on validation data to achieve a desired trade-off between false positives and false negatives (S, 2023).

### 2.2 CURRENT APPROACHES

Modern face recognition models predominantly rely on deep convolutional networks trained with angular margin losses. Early approaches like the triplet loss (Ming et al., 2017) have given way to additive angular margin methods such as SphereFace (Liu et al., 2018), CosFace (Wang et al., 2018), and ArcFace (Deng et al., 2022). These methods enforce a hyperspherical embedding space

with clear angular margins between identities. For example, ArcFace introduces an additive angular margin penalty to the softmax loss, $L = -\frac{1}{N} \sum_{i=1}^{N} \log \frac{e^{s \cos(\theta_{y_i} + m)}}{e^{s \cos(\theta_{y_i} + m)} + \sum_{j \neq y_i} e^{s \cos \theta_j}}$. Where $\theta_{y_i}$ is the angle between an embedding and its class center, $m$ is the margin, and $s$ is a scaling factor. This formulation enhances inter-class separability, leading to robust matching performance. Since the training objective is designed solely to maximize discriminative margins rather than probability estimation additional post-processing calibration steps are required to transform embedding-level similarity scores into reliable probability estimates.

## 2.3 POST-PROCESSING CALIBRATION AND FAIRNESS

We focus on post-processing methods that adjust trained models' outputs to improve fairness and accuracy (Salvador et al., 2022; Kotwal & Marcel, 2025). These methods use calibration/validation sets, common components of facial recognition datasets, making them practical for deployment.

FairCal, the current state-of-the-art, applies K-means clustering to face embeddings forming pseudo-demographic groups, then uses beta calibration within each cluster (Salvador et al., 2022). While effective, it depends on clustering quality and appropriate cluster selection, potentially missing local variations within the embedding space.

Other approaches include: AGENDA which uses adversarial networks to remove demographic information from embeddings (Dhar et al., 2020); PASS employing adversarial formulation with discriminator-based attribute suppression (Dhar et al., 2021); FTC replacing similarity scores with learned fair classifiers (Terhorst et al., 2020); and FSN using K-means clustering with cluster-specific normalization (Terhörst et al., 2020). Methods requiring demographic labels at inference (GST (Robinson et al., 2020), Oracle calibration, DemoNorm (Linghu et al., 2024)) are impractical for deployment. FRAPPÉ (Tifrea et al., 2024) learns additive corrections without requiring sensitive attributes at inference but needs complex fairness penalty tuning.

Building on this work, our approach employs a compact MLP to implicitly learn region-specific mappings from cosine similarity to probabilities, providing greater adaptability than cluster-based methods without manual tuning.

## 3 METHOD

By leveraging the averaged, normalized face embeddings alongside the cosine similarity score—and training on a dedicated calibration set—Adaptive Calibration adapts the calibration function locally. We present two variants of our method: Adaptive Calibration, which employs an efficient MLP to learn non-linear relationships between embedding contexts and calibrated outputs, and Adaptive Calibration Linear, which runs logistic regression on embedding based features. In practice, many real-world training pipelines provide multiple calibration sets (e.g., validation and holdout sets), which makes these practical tools for practitioners (Salvador et al., 2022). The model is trained using leave one out cross validation. where the training data in each iteration consists exclusively of the existing pairs from the designated training folds, without artificially generating additional pairs from metadata.

## 3.1 MLP-BASED CALIBRATION

We employ a small MLP to map the concatenation of the averaged face embedding (computed as $\frac{z_1 + z_2}{2}$ and then L2-normalized), and the cosine similarity score, to a calibrated probability. Denote this network as $g_\phi$, with $p = g_\phi\left(\frac{z_1 + z_2}{2}, s(z_1, z_2)\right)$. Where $s(z_1, z_2)$ is the cosine similarity between embeddings $z_1$ and $z_2$. The MLP is trained on a dedicated calibration set using the standard binary cross-entropy loss, $\mathcal{L}_{\text{BCE}} = -\frac{1}{N} \sum_{i=1}^{N} \left[ y_i \log(p_i) + (1 - y_i) \log(1 - p_i) \right]$. Ensuring that the output probabilities align with the ground-truth labels.

## 3.2 LINEAR CALIBRATION

We train a logistic regression model on the concatenation of the averaged face embedding and the cosine similarity, $p_{\text{initial}} = \sigma\left(w^T \left[\frac{z_1 + z_2}{2}, s(z_1, z_2)\right] + b\right)$. Where $\sigma$ is the sigmoid function, $w$ is a

learned weight vector, and $b$ is a bias term. Notably, for a fixed average embedding, the probability varies monotonically with the cosine similarity, as the sigmoid function preserves the monotonicity of its input. The logistic regression model is trained by minimizing the binary cross-entropy on a calibration dataset.

### 3.3 IMPLEMENTATION DETAILS

Our MLP version of Adaptive Calibration consists of an MLP with an input layer of size (embed_dim + 1), one hidden layer with ReLU activation, and an output layer with sigmoid activation. The network is trained using Adam optimization with a learning rate of $1 \times 10^{-3}$ for 5 epochs, minimizing the binary cross-entropy loss. While we experimented with more complex training pipelines, observed performance improvements were minimal. During inference, embeddings are averaged and normalized; concatenated features are then processed through the MLP to obtain probability estimates.

## 4 EXPERIMENTAL SETUP

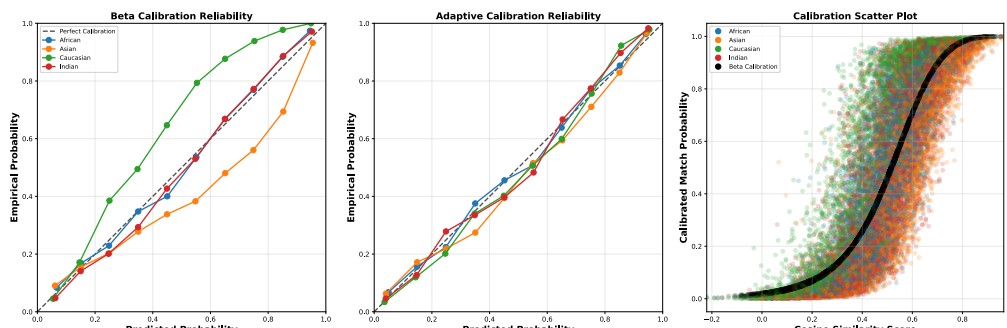

Figure 3: Adaptive Calibration and beta calibration on RFW (FaceNet): Left: reliability diagram comparing predicted probabilities per ethnicity against empirical probabilities for Beta calibration. Center: reliability diagram for Adaptive Calibration. Right: cosine similarity vs match probability score for beta calibration and Adaptive Calibration. Looking at the left and center plots we find that Beta calibration (and consequentially standard thresholding) routinely overestimate the probability of an Asian match being correct, while underestimating the probability of a Caucasian match being correct. This substantially reduces the probability of a Caucasian false positive, relative to Asian, at any threshold. However, the wide spread of overlapping values of each group in the lefthand plot make it clear that a per group adjustment is insufficient.

We evaluate face verification tasks as is standard for post-processing calibration methods. In all experiments, we follow the standard leave-one-out cross-validation protocol analogous to FairCal (Salvador et al., 2022), where calibration parameters are learned exclusively on the training split of each fold and then applied to the held-out test split.

### 4.1 DATASETS AND ANNOTATIONS

The Racial Faces in the Wild (RFW) dataset (Wang et al., 2019) provides face pairs partitioned into four ethnic groups (African, Asian, Caucasian, and Indian), enabling direct measurement of fairness using ground-truth demographic labels. It is partitioned into 10 folds over 24,000 pairs. RFW has become a widely adopted benchmark for evaluating fairness in facial recognition systems across various methods (Salvador et al., 2022).

Previous works(Salvador et al., 2022; Dhar et al., 2021; Linghu et al., 2024) utilized additional datasets BFW and IJB-C, (Robinson et al., 2020; Maze et al., 2018) for similar experiments to ours. However, these are no longer publicly available from official sources, and we omit them from our analysis. Instead, focus on RFW(Wang et al., 2019) and adapt DemogPairs (Hupont & Fernandez, 2019). DemogPairs is evenly split by gender and ethnicity over: Asian, Black and White individuals. While the original dataset was designed to evaluate fairness by comparing state-of-the-art models

across demographic groups, we re-processed DemogPairs for calibration testing. We parsed metadata files to extract both race and gender information for each image and generated all possible image pairs—restricting negative pairs to images within the same demographic groups. Identities were then partitioned into 10 balanced folds using a greedy algorithm that evenly distributes images while keeping subjects in each fold unique, and pairs within each fold were balanced to contain equal numbers of positive and negative examples over 183,600 total pairs. The resulting test pairs and fold assignments will be made public.

## 4.2 Evaluation Protocols

In our evaluation, we use pairwise cosine similarities between face embeddings extracted from the evaluation datasets, and then use calibrated scores—obtained via Adaptive Calibration or other methods—to determine match decisions by applying a threshold. Accuracy is measured by reporting the AUROC and the TPR at fixed FARs of 0.1% and 1.0%, which provide insight into how well the system correctly identifies genuine matches. The predictive equity (PE) gap, captures the difference in TPR between demographic groups, with measurements provided at both 0.1% and 1.0% FAR to quantify fairness in the matching process. This is relevant in policing as a greater PE translates to a greater risk of harm to minority groups (Salvador et al., 2022).

Two complementary area-under-the-curve (AUC) metrics are used to evaluate fairness over all operating points. The first metric, difference AUC FPR, is obtained by considering the global FPR against the maximum FNR observed across all demographic groups. The second metric, AUC FNR, is derived by considering the global FNR against the maximum FPR across groups. Together, these metrics provide a view of how each method balances error rates across different demographic groups over all thresholds rather than at just a few specific points. A lower difference in AUC indicates a fairer model as it suggests that the error rates are more evenly balanced across groups.

To assess whether fairness improvements come at the cost of "leveling down", we calculate the per-group AUROC value and identify the largest number $K$ of the worst-performing groups in a baseline—ranked by their AUROC scores—that all show improvement in AUROC when a new method is applied. The Levels Down (LD) score is then expressed as $\frac{K}{N}$, where $N$ represents the total number of demographic groups. Note that for Oracle, GST and DemoNorm since each demographic group gets its own monotonic transformation, the ranking of scores within each group remains identical to the baseline as such we set this metric to N/A for these methods.

We report average performance across folds, and the LD analysis is based on these averages. We also review the probability adjustment between Adaptive Calibration and beta calibration to provide an intuitive explanation for why it works. Our supplementary materials report the standard deviation per metric across folds and additional metrics.

## 4.3 Pretrained Face Recognition Models

We employ five pretrained face recognition models as fixed feature extractors. Two Facenet models. Facenet (Webface) with am Inception-ResNet-v1 backbone, trained on CASIA-Webface (0.5M images of 10,575 subjects). Facenet (VGGFace2) with an Inception-ResNet-v1 backbone, trained on VGGFace2 (3.31M images of 9,131 subjects) (Schroff et al., 2015). Both Facenet models utilize the weights provided by the standard repository based on the initial method (Sandberg, 2018). We use two models train with ArcFace loss. ArcFace (VGGFace2) obtained from MoZuMa which uses the official ArcFace implementation from InsightFace (Guo & Deng, 2025). From InsightFace we utilized their ArcFace model with ResNet-100 backbone, trained on WebFace600k (0.6M images of 10,000 subjects). And we utilize a GhostFaceNet from DeepFace trained on MS-Celeb-1M (10M images of 100,000 subjects)(Serengil & Ozpinar, 2024).

These models are used as feature extractors in our pipeline. We deliberately selected models representing a range of skill levels and training data diversity to demonstrate the robustness of our calibration approaches across varying baseline performance characteristics. Only GhostFaceNet is evaluated on DemogPairs (image alignment handled by RetinaFace) (Serengil & Ozpinar, 2024; Deng et al., 2019)) as the other models are trained on images in the dataset. The inverse is true for RFW.

## 4.4 BASELINES AND COMPARISONS

We evaluate Adaptive Calibration against state-of-the-art calibration and bias mitigation methods, selecting all that have demonstrated strong fairness performance and/or accuracy in prior work, as well as an unadjusted baseline. We also include FRAPPÉ as it has not previously been tested in this domain and is proposed as a universal fairness calibration method. For methods that do not inherently produce probabilities, we apply beta calibration as is standard in such evaluation (Salvador et al., 2022).

For each calibration method considered, we use the recommended parameters from each paper. Beta calibration serves as our baseline approach for score adjustment. This is effectively the same as the base model outputs. FairCal is applied using $K = 100$ clusters, where beta calibration is performed within each cluster to refine fairness across groups. FSN is implemented with $K = 100$ clusters and targets a False Positive Rate (FPR) of 0.05. GST at a target FPR of 0.05. Oracle Calibration using all available demographic info. FRAPPÉ employing hierarchical clustering with $K = 20$ clusters, adversarial training with a regularization parameter $\lambda = 1.0$, and a learning rate of 5e-5. For DemoNorm we utilize the M1.1 method. For FRAPPÉ the fairness loss is computed as the mean absolute difference between predicted scores across all pairs of demographic groups, defined using provided ethnicity labels in training.

We additionally evaluate Adaptive Calibration on other modern face recognition architectures including AdaFace and MagFace, with extended verification benchmarks including LFW, AgeDB-30, and CA-LFW, demonstrating consistent improvements across diverse model architectures and datasets in the appendix.

Table 1: Per-Group AUROC Values for GhostFaceNet (MS-Celeb-1M) on DemogParis

| Method | African Female | African Male | Asian Female | Asian Male | Caucasian Female | Caucasian Male | Average | Min AUROC |
|---|---|---|---|---|---|---|---|---|
| Baseline | 83.71 | 81.50 | 83.69 | 86.22 | 85.01 | 84.35 | 84.08 | 81.50 |
| FairCal | 85.40 | 82.90 | 84.61 | 87.66 | 87.03 | 86.60 | 85.70 | 82.90 |
| Oracle | 83.71 | 81.50 | 83.69 | 86.22 | 85.01 | 84.35 | 84.08 | 81.50 |
| GST | 83.71 | 81.50 | 83.69 | 86.22 | 85.01 | 84.35 | 84.08 | 81.50 |
| FSN | 80.80 | 79.34 | 81.89 | 83.54 | 82.24 | 81.22 | 81.51 | 79.34 |
| FRAPPÉ | 83.58 | 81.61 | 83.43 | 86.18 | 85.09 | 84.35 | 84.04 | 81.61 |
| DemoNorm (M1.1) | 83.71 | 81.50 | 83.69 | 86.22 | 85.01 | 84.35 | 84.08 | 81.50 |
| Adaptive Calibration (MLP) | **86.99** | **83.92** | **86.11** | **89.85** | **88.70** | **89.69** | **87.54** | **83.92** |
| Adaptive Calibration (Linear) | 84.84 | 81.87 | 84.62 | 87.60 | 86.10 | 86.16 | 85.20 | 81.87 |

*Note: Bold values indicate the highest AUROC in each column. Values in green indicate improvement over the*

*baseline, values in red indicate degradation.*

## 4.5 RESULTS

Figure 3 illustrates why Adaptive Calibration is effective. The left panel shows that Adaptive Calibration adjusts the probability estimates for each pair based on where they fall in the embedding space. Here we see that the mapping to match probability varies depending on the location within the embedding space, and contrast it with beta calibration which provides a monotonic mapping from cosine similarity to match probability. The right panel in 3 illustrates that this advantage translates to better performance in practice. As Adaptive Calibration's predicted probabilities align more closely with true empirical probabilities across diverse demographic groups. In contrast, beta calibration shows larger discrepancies. This fits well with previous work suggesting that thresholds based on demographic criteria led to both accuracy and fairness gains (Robinson et al., 2020; Linghu et al., 2024).

Table 2 summarizes all experiments; see supplementary materials for an expanded version. Across all base models and datasets we find that Adaptive Calibration (linear and MLP variants) has the best AUROC, AUC FPR, and AUC FNR. Adaptive Calibration is the most consistent for PE. With a stronger base model Adaptive Calibration tends to have better TPR accuracy at both 0.1% and 1% but no method clearly dominates this category. Though as baseline performance increases, Adaptive Calibration tends to achieve greater relative performance across each dataset.

Table 2: Results: Accuracy and Fairness Metrics Across Models/Datasets

| Base | Dataset | Method | AUROC (%)↑ | TPR (0.1%)↑ | TPR (1.0%)↑ | PE Gap (0.1%)↓ | PE Gap (1.0%)↓ | AUC (FPR)↓ | AUC (FNR)↓ | LD↑ |
|---|---|---|---|---|---|---|---|---|---|---|
| FaceNet (VGGFace2) | RFW | Baseline | 89.54±0.51 | 17.42±4.64 | 36.02±3.79 | 0.32±0.03 | 2.09±0.40 | 0.1519±0.0124 | 0.1560±0.0097 | – |
| | | FairCal | 91.74±0.54 | 20.65±7.37 | **45.83±3.90** | 0.34±0.03 | 1.45±0.88 | 0.1031±0.0063 | 0.1031±0.0065 | 3/4 |
| | | Oracle | 90.95±0.53 | 22.73±6.34 | 44.22±3.65 | **0.28±0.02** | 1.57±0.55 | 0.1137±0.0059 | 0.1136±0.0062 | N/A |
| | | GST | 90.79±0.52 | 21.98±7.61 | 44.14±3.81 | 0.39±0.04 | 1.48±0.60 | 0.1226±0.0083 | 0.1103±0.0065 | N/A |
| | | FSN | 91.07±0.55 | **22.84±8.11** | 43.01±4.00 | **0.28±0.02** | 1.57±0.80 | 0.1072±0.0073 | 0.1125±0.0078 | 3/4 |
| | | FRAPPÉ | 90.96±0.56 | 21.78±4.91 | 41.59±3.28 | 0.31±0.02 | 1.76±0.69 | 0.1092±0.0072 | 0.1139±0.0086 | 2/4 |
| | | DemoNorm | 90.57±0.53 | 19.95±6.40 | 43.22±3.71 | 0.28±0.02 | 1.93±0.56 | 0.1312±0.0087 | 0.1097±0.0078 | N/A |
| | | Adaptive Calibration MLP | **91.92±0.64** | 14.27±7.51 | 43.87±4.86 | 0.35±0.01 | 1.40±0.46 | 0.1050±0.0093 | 0.1027±0.0145 | 3/4 |
| | | Adaptive Calibration Linear | 91.76±0.55 | 15.06±5.35 | 44.81±4.39 | 0.35±0.02 | **1.39±0.34** | **0.1013±0.0066** | **0.1023±0.0069** | 3/4 |
| GhostFaceNet (MS-Celeb-1M) | DemogParis | Baseline | 83.87±1.26 | 18.10±3.02 | 35.94±3.81 | 0.25±0.13 | 1.62±0.53 | 0.1983±0.0214 | 0.1967±0.0261 | – |
| | | FairCal | 85.42±1.15 | 28.60±2.61 | 40.51±2.66 | 0.22±0.06 | 1.13±0.32 | 0.1839±0.0196 | 0.1785±0.0236 | 6/6 |
| | | Oracle | 83.66±1.19 | 19.00±2.40 | 35.97±3.35 | **0.17±0.07** | 1.17±0.47 | 0.2072±0.0252 | 0.1933±0.0221 | N/A |
| | | GST | 83.62±1.21 | 19.31±2.62 | 36.32±3.22 | 0.20±0.04 | 1.02±0.48 | 0.2169±0.0276 | 0.1893±0.0181 | N/A |
| | | FSN | 81.31±1.18 | 26.19±2.20 | 39.17±2.37 | 0.26±0.13 | 1.25±0.26 | 0.2346±0.0249 | 0.2112±0.0158 | 0/6 |
| | | FRAPPÉ | 83.92±1.30 | 18.15±2.74 | 35.81±3.36 | 0.33±0.12 | 1.46±0.45 | 0.1997±0.0217 | 0.1893±0.0228 | 2/6 |
| | | DemoNorm | 83.79±1.19 | 19.59±2.24 | 36.24±3.20 | 0.27±0.06 | 1.02±0.51 | 0.2047±0.0233 | 0.1792±0.0158 | N/A |
| | | Adaptive Calibration MLP | **87.37±1.06** | **29.45±2.28** | **42.01±3.08** | 0.20±0.05 | 1.30±0.40 | **0.1624±0.0142** | **0.1637±0.0200** | 6/6 |
| | | Adaptive Calibration Linear | 84.87±1.22 | 12.18±3.60 | 31.20±4.28 | 0.29±0.10 | 1.48±0.41 | 0.1831±0.0142 | 0.1863±0.0263 | 6/6 |
| ArcFace (WebFace600k) | RFW | Baseline | 99.71±0.06 | 87.94±5.54 | 96.58±0.61 | 0.30±0.02 | 2.20±0.40 | 0.0058±0.0021 | 0.0050±0.0007 | – |
| | | FairCal | 99.72±0.06 | 87.96±10.75 | 97.17±0.59 | 0.37±0.02 | 1.54±0.25 | 0.0061±0.0023 | 0.0046±0.0016 | 0/4 |
| | | Oracle | 99.73±0.06 | 89.71±4.33 | **97.27±0.45** | 0.30±0.02 | 1.32±0.54 | 0.0065±0.0021 | 0.0045±0.0010 | N/A |
| | | GST | 99.71±0.07 | 89.85±4.13 | 97.17±0.52 | 0.37±0.04 | 1.14±0.44 | 0.0069±0.0023 | 0.0041±0.0011 | N/A |
| | | FSN | 99.62±0.07 | 74.32±24.57 | 95.26±1.31 | **0.28±0.03** | 1.91±0.58 | 0.0069±0.0015 | 0.0064±0.0023 | 0/4 |
| | | FRAPPÉ | 99.67±0.07 | 89.22±4.26 | 96.06±0.97 | 0.31±0.01 | 2.05±0.68 | 0.0072±0.0023 | 0.0055±0.0014 | 0/4 |
| | | DemoNorm | 99.72±0.06 | 89.64±4.40 | 97.02±0.65 | **0.28±0.03** | **1.10±0.57** | 0.0060±0.0020 | 0.0035±0.0007 | N/A |
| | | Adaptive Calibration MLP | 99.75±0.07 | 88.56±5.99 | 96.98±0.33 | 0.36±0.01 | 1.52±0.45 | 0.0052±0.0018 | 0.0035±0.0010 | 3/4 |
| | | Adaptive Calibration Linear | **99.76±0.06** | 90.41±4.22 | 97.27±0.42 | 0.34±0.00 | 1.71±0.38 | **0.0046±0.0019** | **0.0034±0.0007** | 4/4 |
| FaceNet (CASIA-WebFace) | RFW | Baseline | 84.69±0.42 | 8.18±4.34 | 24.24±2.17 | 0.27±0.02 | 2.76±0.40 | 0.1970±0.0068 | 0.2490±0.0126 | – |
| | | FairCal | 87.21±0.39 | **15.57±4.72** | 33.10±4.12 | 0.37±0.05 | 1.36±0.50 | 0.1569±0.0047 | 0.1542±0.0056 | 4/4 |
| | | Oracle | 86.09±0.36 | 12.49±4.70 | 29.59±2.92 | 0.37±0.03 | 1.80±0.60 | 0.1684±0.0034 | 0.1726±0.0061 | N/A |
| | | GST | 85.92±0.36 | 12.76±4.69 | 29.93±2.72 | **0.26±0.01** | 1.50±0.72 | 0.1810±0.0058 | 0.1634±0.0064 | N/A |
| | | FSN | 86.67±0.37 | 12.68±5.69 | 32.84±3.47 | 0.29±0.03 | 1.38±0.57 | 0.1620±0.0052 | 0.1606±0.0070 | 4/4 |
| | | FRAPPÉ | 86.04±0.41 | 10.88±4.97 | 30.32±1.83 | 0.27±0.04 | 1.32±0.55 | 0.1708±0.0106 | 0.1699±0.0097 | 2/4 |
| | | DemoNorm | 85.70±0.36 | 11.24±4.35 | 29.35±2.38 | 0.31±0.01 | 1.72±0.57 | 0.1932±0.0051 | 0.1597±0.0052 | N/A |
| | | Adaptive Calibration MLP | **89.45±0.47** | 10.33±6.04 | **35.68±2.20** | 0.27±0.03 | **1.03±0.29** | 0.1364±0.0101 | **0.1337±0.0101** | 4/4 |
| | | Adaptive Calibration Linear | 89.10±0.38 | 11.66±5.54 | 34.21±3.06 | 0.27±0.00 | 1.04±0.44 | **0.1344±0.0055** | 0.1338±0.0063 | 4/4 |

*Note: **Bold** values indicate the best performance for each metric. For LD: Oracle, GST, and DemoNorm are set to N/A as each demographic group gets its own monotonic transformation, meaning the ranking of scores within each group remains identical to the baseline.*

Our experiments demonstrate that the choice between the MLP-based variant and the linear variant of Adaptive Calibration can be guided by the size and quality of the available calibration set. The MLP-based approach, with its increased capacity to model non-linear relationships, is especially beneficial when large calibration datasets are available, allowing it to capture complex region-specific variations in the embedding space. On the other hand, the linear variant, proves highly effective when calibration data is more limited or when the base recognition model already achieves strong performance. This adaptability makes Adaptive Calibration a versatile tool that can be tuned to the practical constraints of different deployment scenarios.

Another key advantage of Adaptive Calibration is its ability to avoid leveling down. Our results show that Adaptive Calibration typically improves performance for almost all groups, with a minimal degradation occurring for the best performing group in two of the experiments. The minimal leveling down observed—often is such that only one of the groups is harmed; see supplementary material. In the majority of cases, Adaptive Calibration elevates performance for disadvantaged groups without degrading the performance of the best-performing group, as seen in Table 1.

These findings underscore that Adaptive Calibration offers a practical and effective solution for post-hoc calibration in face recognition systems, achieving state-of-the-art fairness-accuracy trade-offs.

## 5 THEORY

To understand why Adaptive Calibration achieves superior fairness-accuracy trade-offs, we analyze its theoretical properties. We show that our method's use of local embedding context enables approximate within-group calibration without requiring demographic labels, explaining the empirical improvements

A predictor $\hat{p}$ is globally calibrated if $\mathbb{P}[Y = 1 \mid \hat{p} = p] = p$ for all $p \in [0, 1]$ and achieves within-group calibration if $\mathbb{P}[Y = 1 \mid \hat{p} = p, G = g] = p$ for all groups $g \in \mathcal{G}$. Within-group calibration implies global calibration. Our approach maps L2-normalized embeddings $z_1, z_2 \in \mathbb{R}^d$ through a

Table 3: Per-group calibration, DemogPairs (ArcFace). Mean across folds. $\Delta$ is pooled $-$ group (lower is better).

| Group | $\text{BCE}_{\text{pooled}}$ | $\text{BCE}_{\text{group}}$ | $\Delta\text{BCE}$ | $\text{ECE}_{\text{pooled}}$ | $\text{ECE}_{\text{group}}$ | $\Delta\text{ECE}$ |
|---|---|---|---|---|---|---|
| African Female | 0.3950 | 0.4217 | $-0.0267$ | 0.0270 | 0.0267 | $+0.0003$ |
| African Male | 0.4308 | 0.4541 | $-0.0233$ | 0.0288 | 0.0282 | $+0.0006$ |
| Asian Female | 0.5147 | 0.5233 | $-0.0086$ | 0.0308 | 0.0381 | $-0.0073$ |
| Asian Male | 0.4560 | 0.4745 | $-0.0185$ | 0.0285 | 0.0294 | $-0.0009$ |
| Caucasian Female | 0.3499 | 0.3886 | $-0.0388$ | 0.0256 | 0.0384 | $-0.0128$ |
| Caucasian Male | 0.2899 | 0.3337 | $-0.0437$ | 0.0300 | 0.0328 | $-0.0027$ |

calibrator $g_\phi : [-1, 1] \times \mathbb{S}^{d-1} \to [0, 1]$ that takes cosine similarity $s = z_1^T z_2$ and normalized average location $\bar{z} = \text{normalize}((z_1 + z_2)/2)$ to output $\hat{p} = g_\phi(s, \bar{z})$. We train $g_\phi$ by minimizing binary cross-entropy $\mathcal{L}\text{BCE} = -\frac{1}{n} \sum i = 1^n [y_i \log \hat{p}_i + (1 - y_i) \log(1 - \hat{p}_i)]$, a strictly proper scoring rule that uniquely minimizes at true conditional probabilities.

The key insight follows from Singh & Joachims (2018), who showed that expressive, unregularized classifiers achieve per-group optimality on training data—yielding identical loss when trained on a single group versus all data. Our calibrators (MLPs and logistic regression with high-dimensional inputs) are sufficiently expressive to represent distinct calibration functions across the embedding space. When minimizing BCE on the calibration set, these models effectively learn the optimal calibration for each region indexed by $\bar{z}$, achieving approximately per-group calibration without explicit group labels. For example, on *DemogPairs (ArcFace)*, pooled AC-MLP matches or improves each group's calibration loss, with small ECE differences shown in Table 3.

This contrasts with clustering-based methods like FairCal that fit separate monotonic maps per discrete cluster. When clusters contain heterogeneous groups with different optimal calibration curves, no single monotonic function can simultaneously satisfy all groups, leaving irreducible error proportional to within-cluster demographic heterogeneity. Our continuous conditioning on $\bar{z}$ avoids discretization artifacts and cluster boundary discontinuities while implicitly leveraging the demographic structure naturally present in face embedding spaces (Wang et al., 2019). By allowing smooth adaptation across regions through the average embedding, Adaptive Calibration achieves within-group calibration as a consequence of minimizing a proper loss with sufficient capacity, rather than through explicit fairness constraints that often degrade performance.

# 6 CONCLUSION

In this work, we proposed Adaptive Calibration, a novel postprocessing approach for recalibrating the heuristic based approaches common to facial recognition. The underlying idea of using a secondary classifier to map from distances and location in the embedding space to probabilities is compellingly easy to implement, and leads to the strongest benefits where there is most uncertainty in the results.

The strong results combined with the fact that most of the benefits can be had by directly running logistic regression on top of the average features and cosine similarity between a pair of images makes this work immediately applicable and likely to be of interest to people working in the field as well as researchers. One significant advantage to post-processing methods is that they can be used to adapt existing facial recognition systems to new datasets with different distributions of data, and our extensive experiments show the improvements Adaptive Calibration brings here.

The negative side of post-processing lies in its reliance on additional data, and the performance of Adaptive Calibration is contingent on the availability and representativeness of the calibration data; insufficient or imbalanced calibration sets will limit its effectiveness. Second, while our experiments cover multiple datasets and pretrained models, further evaluations on more diverse and challenging real-world scenarios are necessary to fully validate the approach.

Finally, we note that making an ML system work well for everyone is only one part of ensuring that an overall process is fair. As a concrete example, where police disproportionately deploy facial recognition systems in minority neighborhoods, the burden of false positives leading to an incorrect stop will continue to fall disproportionately on members of minority groups regardless of how accurate we make our algorithms.

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
