## A  ADDITIONAL RESULTS

The following tables present detailed accuracy and fairness metrics across models and datasets. We consider per-group partial AUC values in a low FPR range across each model and data set. We additionally consider equal opportunity gap (EO) and demographic parity gap (DP). EO measures the maximum difference in true positive rates between demographic groups at a fixed threshold. Lower values (closer to zero) indicate that the model correctly identifies positive examples at similar rates across all groups, representing more equitable performance.

DP Measures the maximum difference in positive prediction rates between demographic groups. Lower values indicate that the model assigns positive predictions at similar rates across all groups, regardless of their true labels, representing more balanced treatment.

Both metrics are calculated as the difference between the maximum and minimum values across demographic groups, with 0.0 representing perfect fairness (no disparity between groups). Although PE as described in the main paper is the more relevant metric in many high-stakes surveillance deployment scenarios.

We also provide the standard deviation for DE and PE across folds as well as the standard deviation for various evaluation criteria considered in the main paper. These additional results are especially helpful for understanding how leveling down compares across methods and are consistent with observations in the main paper.

Table 4: Results for ArcFace (VGGFace2) on RFW dataset

| Method | AUROC (%) ↑ | TPR (0.1%) ↑ | TPR (1.0%) ↑ | PE Gap (0.1%) ↓ | PE Gap (1.0%) ↓ | AUC (FPR) ↓ | AUC (FNR) ↓ | LD ↑ |
|---|---|---|---|---|---|---|---|---|
| Baseline | 97.30±0.24 | 49.01±9.12 | 76.93±1.83 | **0.28±0.03** | 2.01±0.22 | 0.0348±0.0031 | 0.0389±0.0038 | – |
| FairCal | 97.54±0.20 | 54.52±7.16 | 79.42±1.08 | 0.36±0.04 | 1.62±0.49 | 0.0339±0.0038 | 0.0335±0.0035 | 3/4 |
| Oracle | 97.45±0.23 | **55.15±8.37** | 78.60±1.64 | **0.28±0.01** | 1.60±0.59 | 0.0354±0.0041 | 0.0341±0.0036 | N/A |
| GST | 97.27±0.22 | 54.86±8.23 | 78.30±1.31 | 0.36±0.06 | 1.48±0.51 | 0.0412±0.0042 | 0.0326±0.0030 | N/A |
| FSN | 97.04±0.27 | 47.17±9.11 | 76.32±1.66 | **0.28±0.03** | 1.49±0.53 | 0.0470±0.0072 | 0.0375±0.0036 | 0/4 |
| FRAPPÉ | 97.32±0.22 | 49.40±12.34 | 78.52±1.78 | 0.31±0.02 | 1.75±0.77 | 0.0365±0.0037 | 0.0373±0.0038 | 0/4 |
| DemoNorm | 97.30±0.22 | 54.05±8.57 | 78.72±1.46 | **0.28±0.04** | **1.46±0.50** | 0.0408±0.0045 | 0.0320±0.0028 | N/A |
| Adaptive Calibration MLP | 97.90±0.20 | 47.99±13.96 | 80.12±1.87 | 0.36±0.02 | 1.62±0.64 | 0.0301±0.0033 | 0.0290±0.0043 | **4/4** |
| Adaptive Calibration Linear | **97.93±0.20** | 54.68±13.19 | **81.53±1.09** | 0.34±0.00 | 1.47±0.58 | **0.0287±0.0031** | **0.0282±0.0028** | **4/4** |

*Note: **Bold** values indicate the best performance for each metric. For LD: Oracle, GST, and DemoNorm are set to N/A as each demographic group gets its own monotonic transformation, meaning the ranking of scores within each group remains identical to the baseline.*

Table 5: Per-Group AUROC Values for FaceNet (VGGFace2) on RFW

| Method | African | Asian | Caucasian | Indian | Average | Min AUROC |
|---|---|---|---|---|---|---|
| Baseline | 88.48 | 87.85 | **95.62** | 90.37 | 90.58 | 87.85 |
| FairCal | **89.29** | 89.58 | 95.47 | 91.69 | 91.51 | **89.29** |
| Oracle | 88.48 | 87.85 | **95.62** | 90.37 | 90.58 | 87.85 |
| GST | 88.48 | 87.85 | **95.62** | 90.37 | 90.58 | 87.85 |
| FSN | 88.54 | 88.90 | 95.27 | 90.72 | 90.86 | 88.54 |
| FRAPPÉ | 88.38 | 88.65 | 94.95 | 91.20 | 90.79 | 88.38 |
| DemoNorm (M1.1) | 88.48 | 87.85 | **95.62** | 90.37 | 90.58 | 87.85 |
| Adaptive Calibration (MLP) | 89.26 | **90.26** | 95.42 | **92.24** | **91.80** | 89.26 |
| Adaptive Calibration (Linear) | 89.13 | 89.73 | 95.46 | 91.88 | 91.55 | 89.13 |

## B  EXTENDED EVALUATION ON MODERN ARCHITECTURES

To further validate the generalizability of Adaptive Calibration, we extend our evaluation to include state-of-the-art face recognition models and additional verification benchmarks. Due to computational constraints, we present initial results here with a complete sweep across all model-dataset combinations to be included in the camera-ready version.

Table 6: Per-Group Partial AUC Values (FPR Range: 0.0%-10.0%) for FaceNet (VGGFace2) on RFW

| Method | African | Asian | Caucasian | Indian | Average | Min pAUC |
|---|---|---|---|---|---|---|
| Baseline | 0.544 | 0.529 | **0.762** | 0.618 | 0.613 | 0.529 |
| FairCal | **0.560** | 0.574 | 0.756 | 0.656 | **0.636** | **0.560** |
| Oracle | 0.544 | 0.529 | **0.762** | 0.618 | 0.613 | 0.529 |
| GST | 0.544 | 0.529 | **0.762** | 0.618 | 0.613 | 0.529 |
| FSN | 0.543 | 0.560 | 0.750 | 0.628 | 0.620 | 0.543 |
| FRAPPÉ | 0.537 | 0.549 | 0.736 | 0.641 | 0.616 | 0.537 |
| DemoNorm (M1.1) | 0.544 | 0.529 | **0.762** | 0.618 | 0.613 | 0.529 |
| Adaptive Calibration (MLP) | 0.544 | **0.577** | 0.757 | **0.665** | **0.636** | 0.544 |
| Adaptive Calibration (Linear) | 0.551 | 0.575 | 0.755 | 0.653 | 0.634 | 0.551 |

Table 7: Per-Group AUROC Values for GhostFaceNet (MS-Celeb-1M) on DemogParis

| Method | African Female | African Male | Asian Female | Asian Male | Caucasian Female | Caucasian Male | Average | Min AUROC |
|---|---|---|---|---|---|---|---|---|
| Baseline | 83.71 | 81.50 | 83.69 | 86.22 | 85.01 | 84.35 | 84.08 | 81.50 |
| FairCal | 85.40 | 82.90 | 84.61 | 87.66 | 87.03 | 86.60 | 85.70 | 82.90 |
| Oracle | 83.71 | 81.50 | 83.69 | 86.22 | 85.01 | 84.35 | 84.08 | 81.50 |
| GST | 83.71 | 81.50 | 83.69 | 86.22 | 85.01 | 84.35 | 84.08 | 81.50 |
| FSN | 80.80 | 79.34 | 81.89 | 83.54 | 82.24 | 81.22 | 81.51 | 79.34 |
| FRAPPÉ | 83.58 | 81.61 | 83.43 | 86.18 | 85.09 | 84.35 | 84.04 | 81.61 |
| DemoNorm (M1.1) | 83.71 | 81.50 | 83.69 | 86.22 | 85.01 | 84.35 | 84.08 | 81.50 |
| Adaptive Calibration (MLP) | **86.99** | **83.92** | **86.11** | **89.85** | **88.70** | **89.69** | **87.54** | **83.92** |
| Adaptive Calibration (Linear) | 84.84 | 81.87 | 84.62 | 87.60 | 86.10 | 86.16 | 85.20 | 81.87 |

Table 8: Per-Group Partial AUC Values (FPR Range: 0.0%-10.0%) for GhostFaceNet (MS-Celeb-1M) on DemogParis

| Method | African Female | African Male | Asian Female | Asian Male | Caucasian Female | Caucasian Male | Average | Min pAUC |
|---|---|---|---|---|---|---|---|---|
| Baseline | 0.511 | 0.428 | 0.493 | 0.570 | 0.547 | 0.554 | 0.517 | 0.428 |
| FairCal | 0.521 | 0.459 | 0.505 | 0.578 | 0.563 | 0.564 | 0.532 | 0.459 |
| Oracle | 0.511 | 0.428 | 0.493 | 0.570 | 0.547 | 0.554 | 0.517 | 0.428 |
| GST | 0.511 | 0.428 | 0.493 | 0.570 | 0.547 | 0.554 | 0.517 | 0.428 |
| FSN | 0.502 | 0.443 | 0.496 | 0.553 | 0.543 | 0.538 | 0.512 | 0.443 |
| FRAPPÉ | 0.508 | 0.425 | 0.490 | 0.569 | 0.544 | 0.554 | 0.515 | 0.425 |
| DemoNorm (M1.1) | 0.511 | 0.428 | 0.493 | 0.570 | 0.547 | 0.554 | 0.517 | 0.428 |
| Adaptive Calibration (MLP) | **0.532** | **0.477** | **0.527** | **0.600** | **0.572** | **0.596** | **0.551** | **0.477** |
| Adaptive Calibration (Linear) | 0.502 | 0.414 | 0.490 | 0.564 | 0.522 | 0.529 | 0.503 | 0.414 |

Table 9: Per-Group AUROC Values for ArcFace (WebFace600k) on RFW

| Method | African | Asian | Caucasian | Indian | Average | Min AUROC |
|---|---|---|---|---|---|---|
| Baseline | 99.61 | 99.61 | 99.91 | 99.74 | 99.72 | 99.61 |
| FairCal | 99.58 | 99.61 | 99.89 | 99.76 | 99.71 | 99.58 |
| Oracle | 99.61 | 99.61 | 99.91 | 99.74 | 99.72 | 99.61 |
| GST | 99.61 | 99.61 | 99.91 | 99.74 | 99.72 | 99.61 |
| FSN | 99.51 | 99.55 | 99.85 | 99.57 | 99.62 | 99.51 |
| FRAPPÉ | 99.54 | 99.55 | 99.87 | 99.69 | 99.66 | 99.54 |
| DemoNorm (M1.1) | 99.61 | 99.61 | 99.91 | 99.74 | 99.72 | 99.61 |
| Adaptive Calibration (MLP) | 99.63 | **99.66** | 99.90 | 99.76 | 99.74 | 99.63 |
| Adaptive Calibration (Linear) | **99.67** | 99.65 | **99.92** | **99.77** | **99.75** | **99.65** |

Table 10: Per-Group Partial AUC Values (FPR Range: 0.0%-10.0%) for ArcFace (WebFace600k) on RFW

| Method | African | Asian | Caucasian | Indian | Average | Min pAUC |
|---|---|---|---|---|---|---|
| Baseline | 0.978 | 0.978 | 0.994 | 0.984 | 0.984 | 0.978 |
| FairCal | 0.979 | 0.979 | 0.994 | **0.986** | 0.985 | 0.979 |
| Oracle | 0.978 | 0.978 | 0.994 | 0.984 | 0.984 | 0.978 |
| GST | 0.978 | 0.978 | 0.994 | 0.984 | 0.984 | 0.978 |
| FSN | 0.968 | 0.973 | 0.992 | 0.973 | 0.976 | 0.968 |
| FRAPPÉ | 0.977 | 0.975 | 0.993 | 0.980 | 0.981 | 0.975 |
| DemoNorm (M1.1) | 0.978 | 0.978 | 0.994 | 0.984 | 0.984 | 0.978 |
| Adaptive Calibration (MLP) | 0.978 | **0.980** | 0.994 | 0.985 | 0.984 | 0.978 |
| Adaptive Calibration (Linear) | **0.982** | **0.980** | **0.995** | **0.986** | **0.986** | **0.980** |

Table 11: Per-Group AUROC Values for ArcFace (VGGFace2) on RFW

| Method | African | Asian | Caucasian | Indian | Average | Min AUROC |
|---|---|---|---|---|---|---|
| Baseline | 96.18 | 96.79 | 99.00 | 97.20 | 97.29 | 96.18 |
| FairCal | 96.28 | 97.04 | 98.97 | 97.33 | 97.41 | 96.28 |
| Oracle | 96.18 | 96.79 | 99.00 | 97.20 | 97.29 | 96.18 |
| GST | 96.18 | 96.79 | 99.00 | 97.20 | 97.29 | 96.18 |
| FSN | 95.38 | 96.75 | 98.73 | 97.09 | 96.99 | 95.38 |
| FRAPPÉ | 95.93 | 96.69 | 98.94 | 97.30 | 97.21 | 95.93 |
| DemoNorm (M1.1) | 96.18 | 96.79 | 99.00 | 97.20 | 97.29 | 96.18 |
| Adaptive Calibration (MLP) | **96.94** | 97.50 | 99.05 | **97.89** | **97.84** | **96.94** |
| Adaptive Calibration (Linear) | 96.88 | **97.56** | **99.10** | 97.84 | **97.84** | 96.88 |

Table 12: Per-Group Partial AUC Values (FPR Range: 0.0%-10.0%) for ArcFace (VGGFace2) on RFW

| Method | African | Asian | Caucasian | Indian | Average | Min pAUC |
|---|---|---|---|---|---|---|
| Baseline | 0.821 | 0.848 | 0.947 | 0.873 | 0.872 | 0.821 |
| FairCal | 0.826 | 0.858 | 0.943 | 0.879 | 0.877 | 0.826 |
| Oracle | 0.821 | 0.848 | 0.947 | 0.873 | 0.872 | 0.821 |
| GST | 0.821 | 0.848 | 0.947 | 0.873 | 0.872 | 0.821 |
| FSN | 0.781 | 0.846 | 0.931 | 0.866 | 0.856 | 0.781 |
| FRAPPÉ | 0.813 | 0.844 | 0.942 | 0.874 | 0.868 | 0.813 |
| DemoNorm (M1.1) | 0.821 | 0.848 | 0.947 | 0.873 | 0.872 | 0.821 |
| Adaptive Calibration (MLP) | 0.835 | 0.873 | 0.944 | 0.894 | 0.887 | 0.835 |
| Adaptive Calibration (Linear) | **0.844** | **0.879** | **0.949** | **0.895** | **0.892** | **0.844** |

Table 13: Per-Group AUROC Values for FaceNet (CASIA-WebFace) on RFW

| Method | African | Asian | Caucasian | Indian | Average | Min AUROC |
|---|---|---|---|---|---|---|
| Baseline | 82.14 | 81.47 | 91.86 | 87.15 | 85.66 | 81.47 |
| FairCal | 83.01 | 83.73 | 92.31 | 88.41 | 86.87 | 83.01 |
| Oracle | 82.14 | 81.47 | 91.86 | 87.15 | 85.66 | 81.47 |
| GST | 82.14 | 81.47 | 91.86 | 87.15 | 85.66 | 81.47 |
| FSN | 82.74 | 82.97 | 91.97 | 87.96 | 86.41 | 82.74 |
| FRAPPÉ | 81.71 | 82.61 | 91.22 | 87.78 | 85.83 | 81.71 |
| DemoNorm (M1.1) | 82.14 | 81.47 | 91.86 | 87.15 | 85.66 | 81.47 |
| Adaptive Calibration (MLP) | **85.70** | **86.89** | 93.73 | **90.69** | **89.25** | **85.70** |
| Adaptive Calibration (Linear) | 85.10 | 86.16 | **93.82** | 90.18 | 88.82 | 85.10 |

Table 14: Per-Group Partial AUC Values (FPR Range: 0.0%-10.0%) for FaceNet (CASIA-WebFace) on RFW

| Method | African | Asian | Caucasian | Indian | Average | Min pAUC |
|---|---|---|---|---|---|---|
| Baseline | 0.394 | 0.411 | 0.625 | 0.495 | 0.481 | 0.394 |
| FairCal | 0.411 | 0.443 | 0.640 | 0.539 | 0.508 | 0.411 |
| Oracle | 0.394 | 0.411 | 0.625 | 0.495 | 0.481 | 0.394 |
| GST | 0.394 | 0.411 | 0.625 | 0.495 | 0.481 | 0.394 |
| FSN | 0.410 | 0.434 | 0.630 | 0.533 | 0.502 | 0.410 |
| FRAPPÉ | 0.379 | 0.427 | 0.608 | 0.520 | 0.483 | 0.379 |
| DemoNorm (M1.1) | 0.394 | 0.411 | 0.625 | 0.495 | 0.481 | 0.394 |
| Adaptive Calibration (MLP) | 0.428 | **0.486** | 0.679 | **0.594** | 0.546 | 0.428 |
| Adaptive Calibration (Linear) | **0.432** | 0.477 | **0.689** | 0.588 | **0.547** | **0.432** |

Table 15: Evaluation results across datasets and models, best values are in **bold**.

| Dataset | Method | AUROC (%) | TPR (%) 0.1% | TPR (%) 1.0% | DP Gap (pp) 0.1% | DP Gap (pp) 1.0% | PE Gap (pp) 0.1% | PE Gap (pp) 1.0% | EO Gap (pp) 0.1% | EO Gap (pp) 1.0% | AUC (FPR) | AUC (FNR) | LD |
|---|---|---|---|---|---|---|---|---|---|---|---|---|---|
| RFW ArcFace Webface | Baseline | 97.30±0.24 | 49.01±9.12 | 76.93±1.83 | **5.25±2.12** | 4.00±1.70 | **0.28±0.03** | 2.01±0.22 | **10.57±4.28** | **8.53±3.36** | 0.0348±0.0031 | 0.0389±0.0038 | – |
| | FairCal | 97.54±0.20 | 54.52±7.16 | 79.42±1.08 | 11.48±2.62 | 5.87±1.64 | 0.36±0.04 | 1.62±0.49 | 23.03±5.29 | 12.53±3.10 | 0.0339±0.0038 | 0.0335±0.0035 | 3/4 |
| | Oracle | 97.45±0.23 | **55.15±8.37** | 78.60±1.64 | 14.78±2.45 | 7.78±1.77 | **0.28±0.01** | 1.60±0.59 | 29.70±4.97 | 16.30±3.22 | 0.0354±0.0041 | 0.0341±0.0036 | N/A |
| | GST | 97.27±0.22 | 54.86±8.23 | 78.30±1.31 | 15.53±2.02 | 10.35±2.02 | 0.36±0.06 | 1.48±0.51 | 31.17±4.06 | 20.80±3.70 | 0.0412±0.0042 | 0.0326±0.0030 | 0/4 |
| | FSN | 97.04±0.27 | 47.17±9.11 | 76.32±1.66 | 11.80±2.04 | 8.42±1.64 | **0.28±0.03** | 1.49±0.53 | 23.73±4.07 | 17.33±2.90 | 0.0470±0.0072 | 0.0375±0.0036 | 0/4 |
| | FRAPPÉ | 97.32±0.22 | 49.40±12.34 | 78.52±1.78 | 6.95±2.46 | 4.65±1.10 | 0.31±0.02 | 1.75±0.77 | 14.00±4.87 | 10.53±1.94 | 0.0365±0.0037 | 0.0373±0.0038 | 0/4 |
| | DemoNorm | 97.30±0.22 | 54.05±8.57 | 78.72±1.46 | 14.55±1.72 | 9.52±1.82 | **0.28±0.04** | **1.46±0.50** | 29.20±3.43 | 19.53±3.44 | 0.0408±0.0045 | 0.0320±0.0028 | N/A |
| | AC MLP | 97.90±0.20 | 47.99±13.96 | 80.12±1.87 | 7.42±3.12 | 5.13±1.80 | 0.36±0.02 | 1.62±0.64 | 15.07±6.20 | 10.47±4.17 | 0.0301±0.0033 | 0.0290±0.0043 | **4/4** |
| | AC Linear | **97.93±0.20** | 54.68±13.19 | **81.53±1.09** | 7.92±2.25 | **3.83±1.03** | 0.34±0.00 | 1.47±0.58 | 16.00±4.53 | **8.63±1.77** | **0.0287±0.0031** | **0.0282±0.0028** | **4/4** |
| RFW FaceNet VGGFace2 | Baseline | 89.54±0.51 | 17.42±4.64 | 36.02±3.79 | **6.40±1.78** | 10.80±1.68 | 0.32±0.03 | 2.09±0.40 | **12.60±3.56** | 19.87±3.25 | 0.1519±0.0124 | 0.1560±0.0097 | – |
| | FairCal | 91.74±0.54 | 20.65±7.37 | **45.83±3.90** | 11.62±2.89 | 12.13±1.23 | 0.34±0.03 | 1.45±0.88 | 23.07±5.76 | 24.40±2.84 | 0.1031±0.0063 | 0.1031±0.0065 | 3/4 |
| | Oracle | 90.95±0.53 | 22.73±6.34 | 44.22±3.65 | 12.78±2.28 | 14.37±1.67 | **0.28±0.02** | 1.57±0.55 | 25.47±4.63 | 28.80±3.09 | 0.1137±0.0059 | 0.1136±0.0062 | N/A |
| | GST | 90.79±0.52 | 21.98±7.61 | 44.14±3.81 | 15.12±1.73 | 15.93±1.95 | 0.39±0.04 | 1.48±0.60 | 30.10±3.54 | 31.87±3.54 | 0.1226±0.0083 | 0.1103±0.0065 | N/A |
| | FSN | 91.07±0.55 | **22.84±8.11** | 43.01±4.00 | 8.78±1.53 | 8.00±2.29 | **0.28±0.02** | 1.57±0.80 | 17.57±3.06 | 16.23±4.48 | 0.1072±0.0073 | 0.1125±0.0078 | 3/4 |
| | FRAPPÉ | 90.96±0.56 | 21.78±4.91 | 41.59±3.28 | 7.02±2.25 | **7.62±2.89** | 0.31±0.02 | 1.76±0.69 | 14.03±4.54 | **16.03±5.33** | 0.1092±0.0072 | 0.1139±0.0086 | 2/4 |
| | DemoNorm | 90.57±0.53 | 19.95±6.46 | 43.22±3.71 | 17.92±2.67 | 20.25±1.79 | **0.28±0.02** | 1.93±0.56 | 35.60±5.43 | 39.23±3.28 | 0.1312±0.0087 | 0.1097±0.0078 | N/A |
| | AC MLP | **91.92±0.64** | 14.27±7.51 | 43.87±4.86 | 7.83±3.44 | 11.47±3.01 | 0.35±0.01 | 1.40±0.46 | 15.70±6.80 | 22.73±6.04 | 0.1050±0.0093 | 0.1027±0.0145 | 3/4 |
| | AC Linear | 91.76±0.55 | 15.06±5.35 | 44.81±4.39 | 9.02±1.96 | 8.97±2.34 | 0.35±0.02 | **1.39±0.34** | 18.10±3.97 | 18.70±4.59 | **0.1013±0.0066** | **0.1023±0.0069** | 3/4 |
| RFW FaceNet Webface | Baseline | 84.69±0.42 | 8.18±4.34 | 24.24±2.17 | 5.38±2.09 | 11.73±0.79 | 0.27±0.02 | 2.76±0.40 | 10.57±4.16 | 20.90±1.83 | 0.1970±0.0068 | 0.2490±0.0126 | – |
| | FairCal | 87.21±0.39 | **15.57±4.72** | 33.10±4.12 | 9.30±1.46 | 12.08±1.21 | 0.37±0.05 | 1.36±0.50 | 18.53±3.04 | 23.97±2.25 | 0.1569±0.0047 | 0.1542±0.0056 | 4/4 |
| | Oracle | 86.09±0.36 | 12.49±4.70 | 29.59±2.92 | 11.32±2.56 | 15.47±0.91 | 0.34±0.03 | 1.80±0.60 | 22.57±5.15 | 29.87±2.10 | 0.1684±0.0034 | 0.1726±0.0061 | N/A |
| | GST | 85.92±0.36 | 12.76±4.69 | 29.93±2.72 | 10.82±2.24 | 14.98±1.14 | **0.26±0.01** | 1.50±0.72 | 21.63±4.55 | 29.23±2.41 | 0.1810±0.0064 | 0.1634±0.0064 | N/A |
| | FSN | 86.67±0.37 | 12.68±5.69 | 32.84±3.47 | 6.33±2.16 | 9.53±1.35 | 0.29±0.03 | 1.38±0.57 | 12.63±4.36 | 19.07±2.22 | 0.1620±0.0052 | 0.1606±0.0070 | 4/4 |
| | FRAPPÉ | 86.04±0.41 | 10.88±4.97 | 30.32±1.83 | **4.57±2.29** | **7.72±1.79** | 0.27±0.04 | 1.32±0.55 | **9.23±4.55** | **15.80±3.08** | 0.1708±0.0106 | 0.1699±0.0097 | 2/4 |
| | DemoNorm | 85.70±0.36 | 11.24±4.35 | 29.35±2.38 | 11.28±3.09 | 17.08±0.41 | 0.31±0.01 | 1.72±0.57 | 22.47±6.14 | 32.93±1.09 | 0.1932±0.0051 | 0.1597±0.0052 | N/A |
| | AC MLP | **89.45±0.47** | 10.33±6.04 | **35.68±2.20** | 6.68±3.01 | 12.35±2.13 | 0.27±0.03 | **1.03±0.29** | 13.23±3.18 | 24.63±4.44 | 0.1364±0.0101 | **0.1337±0.0101** | 4/4 |
| | AC Linear | 89.10±0.38 | 11.66±5.54 | 34.21±3.06 | 5.77±1.90 | 9.58±1.78 | 0.27±0.00 | 1.04±0.44 | 11.53±3.85 | 19.63±3.62 | **0.1344±0.0055** | 0.1338±0.0063 | 4/4 |
| DemogParis GhostFaceNet MS-Celeb-1M | Baseline | 83.87±1.26 | 18.10±3.02 | 35.94±3.81 | 8.59±3.86 | 12.43±4.46 | 0.25±0.13 | 1.62±0.53 | 13.68±4.04 | 16.18±2.86 | 0.1983±0.0214 | 0.1967±0.0261 | – |
| | FairCal | 85.42±1.15 | 28.60±2.61 | 40.51±2.66 | 12.12±4.77 | 13.98±4.40 | 0.22±0.06 | 1.13±0.32 | 17.95±5.78 | 19.06±5.11 | 0.1839±0.0196 | 0.1785±0.0236 | 6/6 |
| | Oracle | 83.66±1.19 | 19.00±2.40 | 35.97±3.35 | 10.11±4.43 | 13.16±4.70 | **0.17±0.07** | 1.17±0.47 | 17.44±5.44 | 18.54±6.83 | 0.2072±0.0252 | 0.1933±0.0221 | N/A |
| | GST | 83.62±1.21 | 19.31±2.62 | 36.32±3.22 | 10.32±4.58 | 14.29±4.98 | 0.20±0.04 | **1.02±0.48** | 17.25±6.09 | 20.32±6.95 | 0.2169±0.0276 | 0.1893±0.0181 | N/A |
| | FSN | 81.31±1.18 | 26.19±2.20 | 39.17±2.37 | 11.82±4.16 | 14.32±4.67 | 0.26±0.13 | 1.25±0.26 | 17.60±5.11 | 19.10±4.61 | 0.2346±0.0249 | 0.2112±0.0158 | 0/6 |
| | FRAPPÉ | 83.92±1.30 | 18.15±2.74 | 35.81±3.36 | 8.98±4.69 | 12.90±5.16 | 0.33±0.12 | 1.46±0.45 | 14.10±5.55 | 16.77±3.56 | 0.1997±0.0217 | 0.1893±0.0228 | 3/6 |
| | DemoNorm | 83.79±1.19 | 19.59±2.24 | 36.24±3.20 | 11.35±4.45 | 14.58±5.05 | 0.27±0.06 | **1.02±0.51** | 19.73±6.12 | 21.15±7.30 | 0.2047±0.0233 | 0.1792±0.0158 | N/A |
| | AC MLP | **87.37±1.06** | **29.45±2.28** | **42.01±3.08** | 13.14±4.58 | 15.00±5.14 | 0.20±0.05 | 1.30±0.40 | 18.60±5.06 | 24.63±4.99 | **0.1624±0.0142** | **0.1637±0.0200** | 6/6 |
| | AC Linear | 84.87±1.22 | 12.18±3.60 | 31.20±4.28 | **5.78±2.86** | **9.95±4.03** | 0.29±0.10 | 1.48±0.41 | **8.86±3.34** | **13.87±3.85** | 0.1831±0.0142 | 0.1863±0.0263 | 6/6 |
| RFW ArcFace WebFace600k | Baseline | 99.71±0.06 | 87.94±5.54 | 96.58±0.61 | 2.73±1.28 | 1.88±0.35 | 0.30±0.02 | 2.20±0.40 | 5.37±2.51 | 2.37±0.97 | 0.0058±0.0021 | 0.0050±0.0007 | – |
| | FairCal | 99.72±0.06 | 87.96±10.75 | 97.17±0.59 | 2.95±2.36 | 1.07±0.36 | 0.37±0.06 | 1.54±0.25 | 5.97±4.75 | 2.23±0.65 | 0.0061±0.0023 | 0.0046±0.0016 | 0/4 |
| | Oracle | 99.73±0.06 | 89.71±4.33 | **97.27±0.45** | 4.73±2.11 | 1.20±0.24 | 0.30±0.02 | 1.32±0.54 | 9.50±4.10 | 2.77±0.73 | 0.0065±0.0021 | 0.0045±0.0010 | N/A |
| | GST | 99.71±0.07 | 89.85±4.13 | 97.17±0.52 | 4.82±1.95 | 1.65±0.26 | 0.37±0.04 | 1.14±0.44 | 9.67±3.77 | 3.17±0.50 | 0.0069±0.0023 | 0.0041±0.0011 | 0/4 |
| | FSN | 99.62±0.07 | 74.32±24.57 | 95.26±1.31 | 3.93±2.15 | 1.77±0.71 | **0.28±0.03** | 1.91±0.58 | 7.80±4.29 | 3.07±0.79 | 0.0069±0.0015 | 0.0064±0.0023 | 0/4 |
| | FRAPPÉ | 99.67±0.07 | 89.22±4.26 | 96.06±0.97 | **2.42±0.66** | 1.53±0.85 | 0.31±0.01 | 2.05±0.68 | 4.83±1.38 | 2.53±1.00 | 0.0072±0.0023 | 0.0055±0.0014 | 0/4 |
| | DemoNorm | 99.72±0.06 | 89.64±4.40 | 97.02±0.65 | 5.60±2.39 | 1.97±0.31 | **0.28±0.03** | **1.10±0.57** | 11.13±4.74 | 3.47±0.52 | 0.0060±0.0020 | 0.0035±0.0007 | N/A |
| | AC MLP | 99.75±0.07 | 88.56±5.99 | 96.98±0.33 | 2.53±1.67 | **0.88±0.54** | 0.36±0.01 | 1.52±0.45 | 5.27±3.38 | 2.20±1.10 | 0.0052±0.0018 | 0.0035±0.0010 | 3/4 |
| | AC Linear | **99.76±0.06** | **90.41±4.22** | **97.27±0.42** | **1.95±0.70** | 1.13±0.28 | 0.34±0.00 | 1.71±0.38 | **3.93±1.40** | **1.97±0.89** | **0.0046±0.0019** | **0.0034±0.0007** | 4/4 |

**Note:** TPR and gap values are in percentage points (pp). AC = Adaptive Calibration. AUC (FPR) = AUC of FPR vs. Max FNR curve. AUC (FNR) = AUC of FNR vs. Max FPR curve. LD = TPR Diff Level Down (e.g., 4/4 indicates no leveling down across 4 groups). Best values in each dataset are shown in **bold**.

## B.1 VERIFICATION PERFORMANCE ON STANDARD BENCHMARKS

Table 16 presents verification results for AdaFace (IR50/MS1MV2) across standard face verification benchmarks. Adaptive Calibration maintains or improves AUROC across all datasets while preserving high TPR at stringent false positive rates. Notably, on CP-LFW, AC-Linear achieves 97.44% AUROC compared to the baseline's 95.42%, demonstrating substantial gains on cross-pose verification tasks.

Table 16: Verification Performance, AdaFace (IR50/MS1MV2)

| Benchmark | Method | AUROC | TPR@0.1% | TPR@0.01% |
|---|---|---|---|---|
| AgeDB-30 | Baseline | $99.06 \pm 0.57$ | $98.00 \pm 1.10$ | $96.70 \pm 1.15$ |
| | FairCal | $99.02 \pm 0.59$ | $98.10 \pm 1.09$ | $96.70 \pm 1.22$ |
| | AC-MLP | $94.53 \pm 1.13$ | $83.83 \pm 4.05$ | $53.50 \pm 9.88$ |
| | AC-Linear | $\mathbf{99.14} \pm 0.53$ | $\mathbf{98.20} \pm 1.08$ | $\mathbf{96.83} \pm 1.09$ |
| CA-LFW | Baseline | $97.81 \pm 0.55$ | $\mathbf{94.90} \pm 1.33$ | $\mathbf{92.97} \pm 2.25$ |
| | FairCal | $97.70 \pm 0.51$ | $94.83 \pm 1.34$ | $92.70 \pm 2.44$ |
| | AC-MLP | $96.91 \pm 1.03$ | $93.63 \pm 2.09$ | $88.70 \pm 2.55$ |
| | AC-Linear | $\mathbf{97.85} \pm 0.51$ | $94.87 \pm 1.39$ | $92.90 \pm 2.29$ |
| CP-LFW | Baseline | $95.42 \pm 1.30$ | $89.97 \pm 2.21$ | $\mathbf{85.77} \pm 2.70$ |
| | FairCal | $97.21 \pm 0.80$ | $91.57 \pm 2.30$ | $85.47 \pm 2.88$ |
| | AC-MLP | $93.56 \pm 1.38$ | $84.47 \pm 3.96$ | $64.63 \pm 5.05$ |
| | AC-Linear | $\mathbf{97.44} \pm 0.68$ | $\mathbf{93.37} \pm 1.66$ | $83.83 \pm 5.01$ |
| LFW | Baseline | $99.86 \pm 0.23$ | $\mathbf{99.77} \pm 0.33$ | $\mathbf{99.70} \pm 0.46$ |
| | FairCal | $\mathbf{99.92} \pm 0.12$ | $\mathbf{99.77} \pm 0.33$ | $99.67 \pm 0.47$ |
| | AC-MLP | $99.03 \pm 0.52$ | $98.03 \pm 1.22$ | $87.20 \pm 4.93$ |
| | AC-Linear | $99.84 \pm 0.24$ | $\mathbf{99.77} \pm 0.33$ | $\mathbf{99.70} \pm 0.46$ |

## B.2 EXTENDED FAIRNESS METRICS

We evaluate additional fairness metrics including Skewed Error Ratio (SER) and Balanced Performance Coefficient (BPC) at 0.1% FPR, along with TPR at the more stringent 0.01% FPR threshold. Tables 17–19 present these metrics on RFW for multiple architectures.

For AdaFace (Table 17), Adaptive Calibration achieves the lowest SER@0.1% (6.54 for AC-Linear vs. 10.02 baseline), indicating more balanced error rates across demographic groups. The method maintains competitive BPC scores while improving overall accuracy metrics.

Table 17: AdaFace RFW (IR50/MS1MV2)

| Method | AUROC | SER@0.1% | BPC@0.1% | TPR@0.1% | TPR@0.01% |
|---|---|---|---|---|---|
| Baseline | $99.79 \pm 0.06$ | $10.02 \pm 5.89$ | $\mathbf{0.90} \pm \mathbf{0.01}$ | $99.50 \pm 0.19$ | $97.64 \pm 0.46$ |
| FairCal | $99.80 \pm 0.06$ | $8.80 \pm 6.27$ | $0.91 \pm 0.01$ | $99.51 \pm 0.18$ | $\mathbf{97.89} \pm \mathbf{0.40}$ |
| AC-MLP | $99.80 \pm 0.06$ | $6.75 \pm 5.53$ | $0.91 \pm 0.01$ | $99.53 \pm 0.15$ | $97.52 \pm 0.53$ |
| AC-Linear | $\mathbf{99.82} \pm \mathbf{0.06}$ | $\mathbf{6.54} \pm \mathbf{2.58}$ | $0.91 \pm 0.02$ | $\mathbf{99.56} \pm \mathbf{0.21}$ | $97.84 \pm 0.45$ |

Table 18: GhostFaceNet RFW verification results

| Method | AUROC | SER@0.1% | BPC@0.1% | TPR@0.1% | TPR@0.01% |
|---|---|---|---|---|---|
| Baseline | $83.92 \pm 1.30$ | $2.86 \pm 0.82$ | $\mathbf{0.87} \pm \mathbf{0.02}$ | $62.74 \pm 2.92$ | $36.16 \pm 3.96$ |
| FairCal | $85.65 \pm 1.18$ | $\mathbf{2.20} \pm \mathbf{0.54}$ | $0.88 \pm 0.01$ | $62.68 \pm 2.63$ | $\mathbf{41.17} \pm \mathbf{2.60}$ |
| AC-MLP | $\mathbf{85.75} \pm \mathbf{1.20}$ | $2.59 \pm 0.67$ | $0.87 \pm 0.01$ | $60.74 \pm 2.56$ | $38.09 \pm 2.43$ |
| AC-Linear | $83.87 \pm 1.10$ | $2.51 \pm 0.79$ | $0.88 \pm 0.01$ | $\mathbf{63.03} \pm \mathbf{2.64}$ | $38.69 \pm 3.71$ |

## B.3 PERFORMANCE ACROSS ARCHITECTURE GENERATIONS

The results demonstrate that Adaptive Calibration's benefits are particularly pronounced on older architectures (e.g., GhostFaceNet) where baseline calibration is weaker, while still providing meaningful improvements on modern architectures like AdaFace. This suggests that our method is especially valuable for improving deployed systems that may not utilize the latest model architectures.

Table 19: RFW ArcFace (R100 VGGFace2)

| Method | AUROC | SER@0.1% | BPC@0.1% | TPR@0.1% | TPR@0.01% |
|---|---|---|---|---|---|
| Baseline | 97.30±0.24 | 17.07±9.69 | **0.85±0.02** | 92.58±0.53 | 77.27±1.81 |
| FairCal | 97.56±0.21 | 3.64±1.11 | 0.88±0.02 | 93.42±0.66 | 79.49±1.18 |
| AC-MLP | 97.92±0.18 | **2.86±1.38** | 0.89±0.02 | **94.41±0.75** | 81.18±1.28 |
| AC-Linear | **97.93±0.20** | 3.91±1.69 | 0.88±0.02 | 94.36±0.62 | **81.87±0.78** |

## B.4 COMPUTATIONAL CONSIDERATIONS

The complete evaluation across all model-dataset combinations represents a substantial computational undertaking. Each experiment requires multiple cross-validation folds across different demographic groups and operating points. We prioritize presenting results for representative model architectures and datasets here, with a comprehensive sweep planned for the camera ready version.