# OpenReview forum: "Adaptive Calibration for Fairer Facial Recognition"
_ICLR.cc/2026/Conference — Submitted to ICLR 2026_

### Official Review · Reviewer_bA9X · 2025-10-28

**Soundness:** 3
**Presentation:** 3
**Contribution:** 2
**Rating:** 4
**Confidence:** 2

**Summary:**

This paper proposes Adaptive Calibration, a postprocessing approach for recalibrating the heuristic based approaches common to facial recognition. The authors leverage a MLP to integrate local embedding context with cosine similarity scores, producing well-calibrated match probabilities. Experiments prove its effectiveness

**Strengths:**

1. The fundamental idea of this paper is technically correct.
2. The paper is well written and easy to follow.
3. The rationale behind each section and the overall motivation are clearly presented and easy to understand.

**Weaknesses:**

1. The core contribution of this work is merely concatenating average embedding positions and cosine similarity into a simple MLP or linear classifier. This "feature concatenation + simple classifier" approach is extremely common in machine learning and lacks technical depth or innovation.
2. The majority of experiments in this paper are conducted on RFW, with limited evaluation on DemogPairs. This raises concerns about the comprehensiveness of the evaluation. What is the rationale behind this imbalanced dataset usage? Conducting extensive experiments across multiple diverse datasets would significantly strengthen the credibility and generalizability of the results.
3. The authors only compare against post-processing methods. How does the proposed approach perform compared to model-level modifications?
4. The method lacks explicit fairness objectives during training, relying solely on binary cross-entropy loss without any fairness constraints. This creates two critical issues: (1) no mechanism ensures the learned calibration function is fair across demographic groups, and (2) minimizing BCE loss does not equate to optimizing fairness metrics.
5. The paper lacks detailed description of how the calibration set is obtained.

**Questions:**

The method essentially combines average embeddings and cosine similarity through a basic MLP, which lacks novelty. The evaluation only compares against post-processing methods while ignoring model-level alternatives. Most critically, the approach lacks explicit fairness objectives.

---

### Official Review · Reviewer_EdhT · 2025-10-31

**Soundness:** 2
**Presentation:** 1
**Contribution:** 3
**Rating:** 2
**Confidence:** 5

**Summary:**

The authors propose a calibration technique that concatenates average embeddings and a cosine similarity score as input to a two-layer MLP trained with standard binary cross-entropy loss.
They show that the output of this network is more calibrated than the raw similarity scores.
Experiments with several pre-trained networks on custom evaluation protocols on well-known benchmark datasets using outdated or custom evaluation metrics show improved performance in fairness.

**Strengths:**

The proposed method is easy, can be applied to any pre-trained face verification network, and does not require predicting demographics.

**Weaknesses:**

1. The authors employ improper evaluation metrics for verification (AUROC, accuracy) and for fairness evaluation (PE, AUC differences, AUC FNR). The authors need to read and understand the new ISO standards (ISO/IEC 19795-10) for fairness evaluation, and report for example Gini at a False Positive Disparities (FPD) of $10^{-3}$.

2. The mathematical notation is unclear.

   a) In line 207, the authors make use of symbols ($z_i$) where they never explain what they should represent.

   b) In line 208, the authors use $s(.,.)$ as the cosine similarity, while s has been used as the arcface scale before. The authors should select a better symbol (such as $\cos$) here.

   c) Line 210 uses symbol $y_i$ without explanation. Hw are the pairs selected to train the network with? Are the two classes balanced somehow?

   d) It is mysterious what $N$ means. In line 210, is seems to be the number of samples (or pairs) in the training dataset, while in line 298, it is defined as the number of demographic groups.

   e) In section 5, the authors repeat equations from section 3, using different nomenclature. They should rather reference earlier equations.


3. The design of the MLP seems counterintuitive.

   a) Concatenating averaged embeddings (with 512 dimensions) and a similarity score (with 1 dimension) without proper calibration leads to a clear overrepresentation of the embedding average, while the score has little impact. An ablation study is required that shows that adding the similarity score improves performance.

   b) Using ReLU as activation function for a two-layer MLP seems inappropriate. ReLU is designed to to fight against vanishing gradients in deep networks, which is not the case here. Other activation functions should be tested in an ablation study.

   c) Similarly, the authors mention to have used "one hidden layer" without providing any intuition on how large this should be. An ablation study showing the impact of this layer size would be helpful.


4. The description of the experimental setup is insufficient.

   a) The authors make use of the "leave-one-out cross-validation" (line 255), but it is left entirely unclear what the "one" is that is left out. One subject, one demographic group? Which demographic groups are used in this work?

   b) In line 262, the authors state that RFW would have ten folds, but the original evaluation protocol defines no such splits. RFW is an evaluation-only dataset, it is not designed to be trained upon. The authors need to train their method on external datasets, and evaluate on the entirety of RFW.

   c) Even for the DemogPairs dataset, the authors developed their own evaluation protocol, which makes their results incomparable to other research.

   d) The description of the "Level Down" evaluation metric is unclear, and it is not understandable why this would be a reasonable fairness metric.

   e) None of the evaluated face recognition networks seems to be state-of-the-art. They are either old (FaceNet), small (ResNet-50) or trained on small datasets (VGGFace2, WebFace600k). There exist larger networks trained on more data, even provided by InsightFace.

   f) The "empirical probability" shown in figure 3 needs to be explained.


5. The authors make use of outdated terminology (such as false rejection rates in line 82). They should use appropriate terms here, as defined by ISO standards (ISO/IEC 19795-8).

6. The authors cite many arXiv papers. They should cite their correct publication venues.

**Questions:**

See weaknesses.

---

### Official Review · Reviewer_EmHe · 2025-10-31

**Soundness:** 1
**Presentation:** 1
**Contribution:** 3
**Rating:** 2
**Confidence:** 4

**Summary:**

The paper proposes a post-processing calibration method to reduce demographic bias in face verification systems. After extracting face embeddings, the authors compute the cosine similarity between two embeddings and concatenate this value with the average of the two embeddings to form the input to a calibration module. Two variants are tested: a multi-layer perceptron (MLP) and a logistic regression model, both producing a final match probability. Experiments are conducted with several backbone face recognition models: FaceNet (trained on VGGFace2 and Casia-WebFace), ArcFace (trained on WebFace600k and VGGFace2), and GhostFaceNet (trained on MS1M); using the RFW and DemogPairs datasets.

**Strengths:**

- The proposed approach is conceptually simple and easy to implement.

- The method could be practical for improving fairness in face recognition systems without retraining the underlying face models.

- The paper focuses on bias mitigation directly at the embedding space level, which is a very interesting direction.

**Weaknesses:**

- The citation to Cherepanova et al. (2022) appears to be inaccurate; I could not find the quoted statement “This imbalance is caused by minority groups occupying low-density regions of the embedding space...” (line 52) in that paper.

- Figure 1 could be clarified; specifically, the meaning of “match probability” and how it differs from “match rate.”

- Details about the training setup for the calibration modules are missing:
  - What dataset is used for training the MLP and logistic regression models?
  - Are embeddings normalized before averaging (would make a difference)?
  - What is the size of the MLP’s hidden layer?

- Figure 2 (left) is illegible; the curves are too close and colors too similar to differentiate.

- The evaluation setup is unclear and does not follow standard practices in face recognition fairness literature.
  - Why are fixed FARs of 0.1% and 1% used, when SOTA models typically report metrics in the 1e-4 to 1e-6 range?
  - The metrics (Global FPR vs. Max Group FNR, Global FPR vs. Group FNR, ..., predictive equity, leveling down) are nonstandard and poorly explained.
  - Without reporting simple performance gaps between demographic groups, it is hard to interpret the fairness results.

- The paper cites many SOTA models in bias mitigation but evaluates primarily on older models (FaceNet) or little known architectures (GhostFaceNet).

- There are frequent references to the beta calibration, but it is not properly explained.

- In, Figure 3, what is meant by “empirical probability,” and how is it computed? A more natural plot would be "empirical probability" in the x-axis and the "predicted probability" in the y-axis. The adaptive calibration appears to be missing in the right plot.

- Section 5 (“Theory”) presents interesting ideas but lacks convincing justification. Averaging two embeddings may place the resulting vector in a region that does not belong to either regions, so it doesn't necessarily give the local context of the embeddings. It is unclear whether concatenating the cosine score adds useful information beyond the embeddings (a 512 vector for ArcFace). An ablation study removing the cosine score input would help validate this.

- AdaFace and MagFace are mentioned briefly (Lines 342–345) without context or inclusion in the main text. Only AdaFace appears in the Supplementary.

**Questions:**

**Suggestions for Improvement**

- Provide clear definitions for all fairness and calibration metrics.

- Include standard bias reporting metrics (e.g., demographic FNR/FPR gaps).

- Clarify training details for the calibration models.

- Conduct ablations on the role of cosine score and embedding averaging.

---

### Official Review · Reviewer_m1FZ · 2025-11-03

**Soundness:** 2
**Presentation:** 3
**Contribution:** 2
**Rating:** 2
**Confidence:** 4

**Summary:**

The paper proposes Adaptive Calibration (AC) as a post-hoc fairness and score calibration technique for face recognition systems.
The method uses cosine similarity and the mean embedding vector of two faces as input to a small  MLP or linear model that outputs calibrated match probabilities. By learning region-specific mappings without demographic labels, the authors claim AC improves both fairness and accuracy compared to prior post-processing methods. The AC is evaluated across several datasets and models reporting improvements in AUROC, fairness AUC metrics, and predictive equity gaps.

**Strengths:**

- Clear motivation and accessible formulation. The motivation of post training solution is valid and clear.
- language is clear and the method is easily reproducable.
- Public release of calibration folds is a positive step for reproducibility.

**Weaknesses:**

- Outdated experimental foundation with heavy reliance on FaceNet (2015) which has no official release and known calibration flaws. This unfortunately makes some conclusions obsolete.
-The proposed approach is a slight variant of FairCal by learning a monotonic mapping conditioned on embeddings rather than clusters. There is no clear theoretical differentiation from prior post-hoc calibration methods. I suggest adding a table showing this theoretical differences more clearly.
- even though FTC mention as a very related solution. and is mentioned that it will be part of the comparison, it is not in the comparison tables and it is not mentioned why is that. Very suspicious reporting.
- Modern architectures (AdaFace, ArcFace R100) are mentioned but only partially evaluated.  most fairness claims stem from legacy models (different unofficially released facenets)
-Improvements are within error margins with no formal statistical tests support claims of superiority

**Questions:**

- Why rely on FaceNet, an outdated model with no official release, for most conclusions?
- Why is FTC cited and discussed but omitted from all experimental comparisons?
- How do the authors justify using non-standard fairness metrics rather than community widely accepted ones?
- Are the improvements over FairCal statistically significant ?
- Do the reported fairness gains persist for modern models (ArcFace-R100, AdaFace) trained on large datasets like WebFace42M?
- Does AC’s MLP variant introduce any risk of overfitting given small calibration sets?
- How does AC perform on fairness metrics that measure disparity at fixed thresholds, rather than integrated AUC values?

---

### Meta-Review · Area_Chair_nTww · 2025-12-06

**Summary:**

Reviewer m1FZ raised the issues: Outdated experimental foundation; The proposed approach is a slight variant of FairCal. No clear theoretical differentiation from prior post-hoc calibration methods. Modern architectures (AdaFace, ArcFace R100) are mentioned but only partially evaluated.

Reviewer EmHe points out the following weaknesses: The citation to Cherepanova et al. (2022) appears to be inaccurate; Details are missing, including the training setup for the calibration modules are missing, dataset used for training the MLP and logistic regression models, and size of the MLP’s hidden layer; Whether embeddings are normalized before averaging (would make a difference) is unclear; Figure 2 (left) is illegible; Evaluation setup is unclear and does not follow standard practices in face recognition fairness literature, together with  many of the others. AdaFace and MagFace are mentioned briefly (Lines 342–345) without context or inclusion in the main text. Only AdaFace appears in the Supplementary.

Reviewer EdhT concerns about the mathematical notation is unclear; None of the evaluated face recognition networks seems to be SOTA. They are either old (FaceNet), small (ResNet-50) or trained on small datasets (VGGFace2, WebFace600k). There exist larger networks trained on more data, even provided by InsightFace, and so on.

Reviewer bA9X notes that the primary contribution of this work is limited to concatenating average embedding positions with cosine similarity and applying them to a basic MLP or linear classifier. Such a “feature concatenation plus simple classifier” strategy is widely used in machine learning and does not demonstrate significant technical novelty or depth.

In sum, all reviewers have many concerns on this paper, and all of them lean toward negative about this paper.
The authors did not provide a rebuttal.

**Reviewer Concerns:**

No rebuttals are provided, so there is no basis to specify which reviewer concerns I believe were addressed by the rebuttal.

**Reviewer Scores:**

No rebuttals have been provided, and the review scores are largely consistent.

---

### Decision · Program_Chairs · 2026-01-26

Reject